# Overexpression of Auxin/Indole-3-Acetic Acid Gene TrIAA27 Enhances Biomass, Drought, and Salt Tolerance in *Arabidopsis thaliana*

**DOI:** 10.3390/plants13192684

**Published:** 2024-09-25

**Authors:** Muhammad Zafar Iqbal, Yuzhou Liang, Muhammad Anwar, Akash Fatima, Muhammad Jawad Hassan, Asif Ali, Qilin Tang, Yan Peng

**Affiliations:** 1College of Grassland Science and Technology, Sichuan Agricultural University, Chengdu 611130, China; m.zafarsindhu@hotmail.com (M.Z.I.);; 2Shandong Laboratory of Advanced Agricultural Sciences at Weifang, Peking University Institute of Advanced Agricultural Sciences, Weifang 261000, China; 3Institute of Tropical Agriculture and Forestry, Hainan University, Haikou 570228, China; 4Institute of Plant Breeding and Biotechnology, MNS University of Agriculture, Multan 60000, Pakistan; 5State Key Laboratory of Crop Gene Exploration and Utilization in Southwest China, Rice Research Institute, Sichuan Agricultural University, Chengdu 611130, China; 6Maize Research Institute, Sichuan Agricultural University, Chengdu 611130, China; tangqilin71@163.com

**Keywords:** auxin/IAA, abiotic stress, *Trifolium repens*, biomass, ROS, antioxidants

## Abstract

White clover (*Trifolium repens* L.) is an important forage and aesthetic plant species, but it is susceptible to drought and heat stress. The phytohormone auxin regulates several aspects of plant development and alleviates the effects of drought stress in plants, including white clover, by involving auxin/indole acetic acid (Aux/IAA) family genes. However, Aux/IAA genes and the underlying mechanism of auxin-mediated drought response remain elusive in white clover. To extend our understanding of the multiple functions of Aux/IAAs, the current study described the characterization of a member of the Aux/IAA family *TrIAA27* of white clover. TrIAA27 protein had conserved the Aux/IAA family domain and shared high sequence similarity with the *IAA27* gene of a closely related species and *Arabidopsis*. Expression of *TrIAA27* was upregulated in response to heavy metal, drought, salt, NO, Ca^2+^, H_2_O_2_, Spm, ABA, and IAA treatments, while downregulated under cold stress in the roots and leaves of white clover. TrIAA27 protein was localized in the nucleus. Constitutive overexpression of *TrIAA27* in *Arabidopsis thaliana* led to enhanced hypocotyl length, root length, plant height, leaf length and width, and fresh and dry weights under optimal and stress conditions. There was Improved photosynthesis activity, chlorophyll content, survival rate, relative water content, endogenous catalase (CAT), and peroxidase (POD) concentration with a significantly lower electrolyte leakage percentage, malondialdehyde (MDA) content, and hydrogen peroxide (H_2_O_2_) concentration in overexpression lines compared to wild-type *Arabidopsis* under drought and salt stress conditions. Exposure to stress conditions resulted in relatively weaker roots and above-ground plant growth inhibition, enhanced endogenous levels of major antioxidant enzymes, which correlated well with lower lipid peroxidation, lower levels of reactive oxygen species, and reduced cell death in overexpression lines. The data of the current study demonstrated that *TrIAA27* is involved in positively regulating plant growth and development and could be considered a potential target gene for further use, including the breeding of white clover for higher biomass with improved root architecture and tolerance to abiotic stress.

## 1. Introduction

*Trifolium repens* L., commonly known as white clover, is an excellent perennial leguminous fodder, widely cultivated due to its high protein content, excellent forage quality, palatability, and biological nitrogen fixation [1,2]. It is extensively used in ornamental lawns and ecological construction due to its creeping growth characteristics and attractive foliage type. Hence, it is an important segment of animal husbandry, soil and water conservation, and urban greening projects. Despite its attractive characteristics, it is vulnerable to drought and heat stress, which reduce its yield and application ranges [2,3]. Phytohormone auxin, specifically auxin/indole-3-acetic acid (IAA), has been proven to enhance drought tolerance in white clover by modulating hormone levels [4,5], gene expression [5], antioxidant defenses [6], and root architecture [7]. Transgenic *Arabidopsis thaliana* (L.) Heynh. with higher endogenous IAA levels showed enhanced drought resistance, while those with lower IAA levels were more susceptible to stress [8]. Auxin signal transduction in plants is mainly controlled by two gene families: auxin/indole-3-acetic acid (Aux/IAA) and auxin response factor (ARF). Of them, Aux/IAA proteins contain a potent transcriptional repression domain that dominates activation domains in ARF transcriptional activators [9], thereby regulating the expression of auxin response genes. 

Aux/IAA proteins are short-lived nuclear proteins that bind with auxin response factors (ARFs) to prevent their activation in the absence of auxin, thereby acting as repressors of ARFs. While at higher auxin levels, AUX/IAA proteins are ubiquitinated via interaction with TRANSPORT INHIBITOR RESPONSE1/AUXIN SIGNALING F-BOX (TIR1/AFB) receptors and degraded by 26S proteasome, thereby releasing ARFs and initiating transcription of downstream auxin-responsive genes [10,11,12]. *Aux/IAA* family genes contain four highly conserved typical domains [13]. Domain I at the N-terminal is a repressor domain containing an ethylene response factor (ERF) associated with amphiphilic repression (EAR) leucine repeats LxLxL that can recruit TOPLESS (TPL) co-repressor [14]. Domain II has a conserved sequence, “GWPPV”, associated with protein stability and can interact with F-BOX proteins SCF^TIR1^ to regulate protein turnover by inducing rapid protein instability and degradation [9]. Domain III has an amphipathic Bαα-fold with functions and structure analogous to the DNA recognition motif [15]. Domain IV possesses an SV40 nuclear localization signal (NLS) “PKKKRKV” and an acidic region. Domains III and IV are mostly involved in the protein–protein interaction and can form homodimers or heterodimers with other proteins by electrostatic interaction [16]. Generally, domain III/IV interacts with the auxin response factor (ARF). However, four domains are not equally found in all plant species; for example, potatoes and tomatoes lack domains I and II [17,18], and papaya lacks domains III and IV [19]. Aux/IAA proteins can interact among themselves or with other proteins in different combinations, thereby increasing the diversity/complexity of auxin signaling for regulating growth and physiology [20,21].

*Aux/IAA* family genes quickly respond to auxin homeostatic changes and play important roles in regulating growth and development in plants [22,23,24]. In *A. thaliana*, Aux/IAA genes may show pleiotropic effects, and a mutation in a single gene can affect the normal growth and development of *A. thalliana* plants [25]. Mutation in the conserved sequence “VGWPPV” of domain II increased protein stability, resulting in failure of SCF^TIR1^ protease complex binding and ubiquitination modification, thus resulting in protein degradation without auxin homeostasis regulation [26,27,28,29]. Similar mutations in *A. thaliana* genes resulted in abnormal hypocotyl elongation and leaf deformation in *AtIAA6*, *AtIAA7*, and *AtIAA17* [30]; loss of radicle growth in *AtIAA12* [31]; cotyledon fusion and reduced root in *AtIAA18* [32]; dominant tap root growth without lateral roots in *AtIAA28* [33], indicating their functioning in different regulatory pathways. Aux/IAA proteins can function independently or in combination with ARFs or may work in a network [34,35,36]. The *AtIAA6*, *AtIAA9*, and *AtIAA17* of *A. thaliana* show additive effects in controlling adventitious root initiation and can interact with ARF6 and ARF8, thereby repressing their functions in adventitious root development [37]. Mutations in *Arabidopsis* genes *AtIAA7* [38], *AtIAA3* [39], *AtIAA8* [40], *AtIAA1* [41,42], *AtIAA28-1*, and *IAA14* [26,43] have also been reported to cause developmental changes in *A. thaliana* plants, suggesting promising roles of Aux/IAA genes in the growth and development of plants. Despite regulating developmental processes, Aux/IAA genes also respond to environmental stresses in plants, such as drought stress in *A. thaliana* [44] and *Brachypodium distachyon* (L.) P. Beauv. [45] and drought and salt stress in *Oryza sativa* L. [46,47]. Drought stress and exogenous application of IAA changed endogenous levels of IAA and caused the differential expression of Aux/IAA genes in white clover as well [4,5,48]. 

Specifically, silencing of *Sl-IAA27* increased auxin sensitivity and root growth while reducing leaf chlorophyll content, ovule and pollen fertility, fruit size, and *Arbuscular mycorrhizal* colonization in tomatoes (*Solanum lycopersicum*) [49,50]. The ethylene-responsive factor of tomatoes “S1-ERF.B3” regulated ethylene and auxin homeostasis by downstream interacting with Sl-IAA27 [51]. Also, *Sl-IAA27* regulated the expression of strigolactone biosynthesis genes *SI-D27* and *SI-MAXI* via modulating the expression of NSPI transcriptional factor [50,52]. Overexpression of the blueberry (*Vaccinium corymbosum*, “Northland”) gene *Vc-IAA27* in *A. thaliana* promoted lateral branches with curled leaves and shorter sterile siliques [53]. Overexpression of apple (*Malus domestica*) *MdIAA27* produced deeper roots with more adventitious roots, thereby improving phosphorus deficiency tolerance in apple [54]. Exogenous application of auxin on white clover improved drought tolerance [48]. Transcriptome analysis of white clover under drought stress revealed differential expression of Aux/IAA genes, including *TrIAA27* with eight-fold higher transcript levels [4], indicating its putative role under drought stress responses. Still, no *TrIAA27* gene has been cloned from white clover nor have the responses of *TrIAA27* been reported under drought and salt stress. Cloning and functional characterization of the white clover gene *TrIAA27* would help to better understand the molecular mechanism of abiotic stress response in white clover. 

## 2. Results

### 2.1. Sequence Characteristics of TrIAA27 

The full gene length of *TrIAA27* was 1459 bp with an open reading frame (ORF) of 912 bp, encoding a protein of 304 amino acids (Figure 1a,c). The predicted molecular mass of *TrIAA27* protein was 32.79 kDa, with a theoretical pI of 7.49, and containing a typical domain of the Aux/IAA family (PF02309), which is rapidly induced by the plant hormone auxin. This specific domain interacts with auxin-responsive factors (ARFs) and inhibits the transcription of genes activated by ARFs. Thus, the *TrIAA27* gene belongs to a large family *Aux/IAA* found in plant species (Figure 1b). A phylogenetic tree constructed based on the neighbor-joining method using TrIAA27 amino acids with closely related species and Aux/IAA family genes of *A. thaliana* showed that TrIAA27 is closely related to IAA27 proteins of *Trifolium pretense* and *Medicago truncatula*. Among the Aux/IAA genes of *A. thaliana*, TrIAA27 forms a distinct clade with AtIAA27, AtIAA9, and AtIAA8 proteins (Figure 1d). These three proteins form a distinct clade (clade B) in plant species [49]. The structural analysis of the TrIAA27 protein showed that it contained a conserved Aux/IAA (auxin/indole-3-acetic acid) domain, which has been reported to be rapidly induced auxin. Some members of this family are longer and contain an N terminal DNA binding domain, but the function of this region is uncertain.

### 2.2. Expression Pattern of White Clover Gene TrIAA27 in Response to Different Stimuli

One-month-old plants of white clover were subjected to exogenous application of heavy metals, abiotic stresses, signaling molecules, and phytohormones treatments for expression pattern analysis of *TrIAA27*. Under heavy metal (CdSO_4_—600 mM) stress, *TrIAA27* significantly upregulated (*p* < 0.05) up to the first 6 h in the roots and leaves and then decreased linearly; however, expression was more obvious in leaves than roots with an increase of 2.6-, 2.4-, 2.9-, 2.4-, and 2.5-fold at 1.5 h, 3 h, 6 h, 12 h, and 24 h, respectively (Figure 2a). An unusual upregulation of *TrIAA27* was observed under cold (4 °C) stress. Expression of *TrIAA27* increased up to 1.5 h in roots, decreased at 3 h, and then again increased at 6 h, reaching a maximum peak with a 1.53-fold increase compared to the control. Except at 6 h, *TrIAA27* expression remained more obvious in leaves than in roots under cold stress. In leaves, expression was significantly upregulated at 1.5 h and 3 h (*p* < 0.05) and then leveled under cold stress (Figure 2b). High temperature (35 °C) linearly downregulated the expression of *TrIAA27* both in roots and leaves at least up to 24 h; however, suppression was stronger in leaves than roots up to the first 6 h, then reached almost a similar level both in roots and leaves (Figure 2c). Under drought stress (15% PEG6000 *w*/*v*), expression of *TrIAA27* was strongly upregulated in leaves, linearly increased up to the first 6 h, and then decreased. The maximum peak was observed at 6 h in leaves at 6.8-fold higher than in the control, while this upregulation remained higher in leaves than in roots at all observed periods (Figure 2d). Under salt stress (200 mM NaCl), *TrIAA27* expression was significantly upregulated in leaves and roots compared to the control. Expression was more obvious in leaves than in roots. Expression was linearly increased up to the first 6 h in the roots (3.2-fold higher) and then decreased, while in leaves, it first increased up to 12 h with a 9.8-fold increase and then decreased (Figure 2e).

The upregulation of *TrIAA27* in response to calcium Ca^2+^ (CaCl_2_—5 mM) and hydrogen peroxide H_2_O_2_ (10 mM) signaling molecules was more obvious in roots compared to leaves. However, *TrIAA27* upregulation in response to nitrous oxide (NO) and spermine (Spm) was stronger in leaves compared to roots. The upregulation of *TrIAA27* by NO treatment was 1.7, 1.7, 2.5, 3.4, and 1.3 folds higher at 1.5 h, 3 h, 6 h, 12 h, and 24 h, respectively, in leaves compared to the control (Figure 2f). The expression of *TrIAA27* in response to Ca^2+^ signaling molecules significantly upregulated in roots showed linear increasing trends up to 6 h (2.5-fold higher than control) and then decreased; however, it remained significantly higher up to 24 h compared to the control. Notably, the expression of *Tr-IAA27* significantly downregulated in leaves (<0.05) in response to Ca^2+^ treatment under all observed time spans (Figure 2g). In response to hydrogen peroxide treatment, the expression of *TrIAA27* was significantly upregulated in roots and was highest at 12 h (3.4 folds). However, upregulation was less obvious in leaves compared to roots (Figure 2h). Exogenous application of spermidine (Spm) significantly upregulated the expression level of *TrIAA27* both in roots and leaves under all observed time spans. The highest peak was observed at 6 h with 1.6- and 2.5-fold increases in roots and leaves, respectively (Figure 2i). The phytohormones (IAA and ABA) significantly upregulated the expression of *TrIAA27* both in roots and leaves at all observed periods, while expression was stronger in leaves compared to roots (Figure 2j,k). After ABA treatment, the highest peak was observed at 6 h with a 4.2-fold increase in roots and at 12 h with a 10.5-fold increase in leaves (Figure 2j). After IAA treatment, *TrIAA27* linearly increased up to 12 h both in roots and leaves, with more obvious effects in leaves than roots. The expression was 4.2-fold higher in roots and 8.6-fold higher in leaves at 12 h after IAA treatment (Figure 2k).

### 2.3. Subcellular Localization of TrIAA27

Using the WoLF PSORT program, the *TrIAA27* protein was predicted to be localized in the nucleus and chloroplast. To detect the subcellular localization of the *TrIAA27* protein, the coding region of *TrIAA27* was combined with the 3′ ends of the GFP gene and driven by the 35S promoter. The GFP gene alone, under the control of the 35S promoter, served as the control. The subcellular localization of *TrIAA27* was determined in a transient expression system in *Nicotiana benthamiana* leaves, and 35S::*TrIAA27*::GFP fusion protein was localized in the nucleus (Figure 3).

### 2.4. Overexpression of TrIAA27 Improves Plant Size and Roots of Transgenic Arabidopsis thaliana Plants

This study also used an overexpression approach to address the physiological significance of the TrIAA27 protein. The *TrIAA27* gene was constructed in the pCAMBIA1300 vector containing CaMV35S promoter and transferred into *A. thaliana* by the *Agrobacterium*-mediated floral dip transformation method. Six independent homozygous lines were developed. Later, two *TrIAA27* overexpression lines (OE3 and OE5) were selected for further experiments based on the relatively higher expression of *TrIAA27* (Figure 2l). The real-time PCR analysis showed that *TrIAA27* overexpressed in transgenic lines with expression levels up to 10-fold higher than the wild-type (Figure 2l). The dark-grown etiolated seedling on ½ MS solid medium manifested different auxin-related phenotypic differences, such as OE lines showing increased apical hook angles under etiolated conditions (Figure 4a–c) and having significantly longer hypocotyls (Figure 4d,i). By germinating on Murashing and Skoog (MS) solid media, overexpression (OE) lines showed a significantly higher number of secondary roots and an increased main root length compared to the wild-type (Figure 4e,j,k). Ten-day-old seedlings grown on ½ MS media were transferred to nutrient soil to observe growth differences between the wild-type and OE lines. The penultimate leaves were used to measure the length and width and photosynthesis-related traits. Whole plants were used before the bolting stage to measure fresh and dry weight, and plant height was measured at maturity. The results showed that plant height, leaf length, leaf width, fresh weight, and dry weight of OE lines were significantly higher compared to the wild-type (Figure 4f–h,l–p), indicating that the overexpression of *TrIAA27* can increase the biomass of *A. thaliana* plants. Total chlorophyll content (a + b), stomatal conductance, water use efficiency, performance index on absorption basis, and net photosynthesis rate were slightly higher in overexpression lines compared to the wild-type, but except for total chlorophyll content, all were statistically non-significant (Appendix A).

### 2.5. Overexpression of TrIAA27 in A. thaliana Improved Drought Stress Tolerance of Transgenic Plants

Drought and salt stress were applied to wild-type and overexpressed transgenic *A. thaliana* on ½ MS medium supplemented with sucrose and agar and in the soil. Five-day-old wild-type and overexpression *A. thaliana* seedlings were transferred onto ½ MS plates containing 100 mML^−1^ or 200 mML^−1^ mannitol. Growth and phenotypic changes were observed daily for up to 10 days. Overexpression lines (OE-3 and OE-5) showed significantly increased root growth compared to the wild-type under 100 mML^−1^ and 200 mML^−1^ mannitol-induced drought-stressed conditions (Figure 5a,c,d). One-week-old overexpression and wild-type *A. thaliana* plants were transferred into the soil and grown for an additional two weeks under optimal growth conditions. Then, drought stress treatment was induced by withholding irrigation. Plants were well-watered once, followed by stopping irrigation until phenotypic difference appeared approximately after 14 days of water withholding. Overexpression lines OE3 and OE5 showed less wilting/drying rates compared to the wild-type (Figure 5b). By rewatering, more than 90% of overexpression *A. thaliana* plants recovered after 3 days compared to the around 60% recovery rate in wild-type *A. thaliana* plants, indicating relatively less damage at the cellular level in overexpression lines compared to wild-type plants. The observations indicated that the overexpression of *TrIAA27* significantly improved drought tolerance in transgenic *A. thaliana* at the seedling and vegetative growth stages. Fresh weight, dry weight, and photochemical efficiency determine how plants grow under stressed conditions, and the plants’ relative water contents (RWC) showed their water status in terms of physiological consequences under water-deficient conditions. The cell membrane permeability and lipid peroxidation are the general indicators of damage in plants caused by different stresses and are measured as electrolyte leakage and malondialdehyde (MDA) contents, respectively. Lipid peroxidation is generally associated with the oxidation of lipids of the cell membrane; thus, its quantification determines the extent of cell membrane damage. The current study determined the phytochemical efficiency and performance index after 10 days of water withholding. Overexpression lines OE3 and OE5 had significantly higher fresh and dry weights, root lengths, performance indices on absorption basis (PI), Fm/Fv ratios, and relative water contents, with significantly reduced electrolyte leakage (%) compared to the wild-type (Figure 5e–j). These physiological and phenotypic observations indicated that the overexpression of *TrIAA27* can improve drought stress tolerance in plants.

### 2.6. Overexpression of TrIAA27 Improved the Salt Stress Resistance of Transgenic Arabidopsis

To observe how *TrIAA27* functions under salt stress conditions, overexpression lines OE3 and OE5 along with wild-type *A. thaliana* were subjected to salt treatment on ½ MS medium supplemented with sucrose and agar and in the soil. Five-day-old *A. thaliana* seedlings were transferred on ½ MS medium supplemented with 100 mML^−1^ or 150 mML^−1^ NaCl. Phenotypic changes were observed daily and used for statistical comparisons after 10 d. The results showed that *TrIAA27* overexpression lines showed a significantly higher survival rate and root growth compared to wild-type *A. thaliana* (Figure 6a,i). Additionally, 10-day-old OE3, OE5, and wild-type *A. thaliana* were also transferred into nutrient soil, raised for an additional two weeks, and subjected to salt stress treatments by irrigating with 100 mML^−1^, 200 mML^−1^, and 300 mML^−1^ NaCl gradient water. Each NaCl concentration was applied twice after every two days. RWC, EC, photochemical efficiency, and the photosynthesis performance index were measured at the seventh d of treatment (after 2 times 100 mM and 2 times 200 mM saline water irrigation), and the survival rate was recorded at the twelfth d of treatment. The results showed that the means of Fv/Fm ratio (0.80, 0.82, and 0.57 for OE3, OE5, and WT, respectively), performance index on an absorption basis (2.02%, 2.48%, and 0.26% for OE3, OE5, and WT, respectively), RWC (75.42%, 77.61, and 65.38% for OE3, OE5, and WT, respectively), chlorophyll contents (3.48 mg, 3.51 mg, and 1.9 mg for OE3, OE5, and WT, respectively), and survival rate (66.22%, 70.33%, and 40.33% for OE3, OE5, and WT, respectively) were significantly higher for overexpression lines compared to the wild-type, while the means of EC (46.66%, 44.03%, and 62.33% for OE3, OE5, and WT, respectively) were significantly lower for overexpression lines compared to wild-type *A. thaliana* (Figure 6b–h), indicating improved salt (NaCl) resistance via *TrIAA27* overexpression in *A. thaliana*.

### 2.7. Physiological Indicators and Enzyme Activity under Drought and Salt Stress in Wild and Overexpression Arabidopsis

The enzymatic activity in the wild-type and *TrIAA27* overexpression lines of *A. thaliana* was analyzed after subjecting them to drought and salt stress in the nutrient soil in controlled conditions. The endogenous contents of two oxidants, “MDA and H_2_O_2_”, and two antioxidants, “CAT and POD”, were measured. The results showed that both transgenic lines overexpressing *TrIAA27* had significantly lower MDA and H_2_O_2_ quantities than wild types under drought and salt stress conditions (Figure 7a,b). Accordingly, both antioxidants (CAT and POD) were found to be significantly higher under drought and salt stress conditions (Figure 7c,d). The elevated levels of the antioxidants and reduced oxidant contents in both overexpression lines compared to the wild-type revealed that overexpression of *TrIAA27* in *A. thaliana* can regulate the quantities of auxin/indole-3-acetic acid under stress conditions, as its regulation has already been reported to enhance the adoption of plant species under drought and salt stress conditions [48,55]. The comparative lower oxidants and high antioxidants in plant leaves show the ability of a genotype to better resist adverse environmental conditions. Thus, based on phenotypic, physiological, and enzymatic indicators, it is evident that the *TrIAA27* of white clover positively regulates the drought and salt stress in *Arabidopsis*.

## 3. Discussion

In this study, we identified a theretofore uncharacterized white clover gene and designated it *TrIAA27* (a member of the auxin/indole-3-acetic acid family), which encodes IAA domain-containing protein. Exogenous application of auxin/indole-3-acetic acid (IAA) induced the expression of *TrIAA27* in the leaves and roots of white clover (Figure 2j), consistent with findings that *Aux/IAA* family genes quickly respond to auxin homeostatic changes in plants [56,57], including *TrIAA27* [49]. Unusually, the highest expression of *TrIAA27* was observed in the roots of white clover at 6h under cold stress. In rice roots, IAA levels decreased under drought stress but considerably increased under cold and heat stress. Transcript levels of many genes involved in the IAA biosynthesis and signaling changed under these stresses and were essentially in agreement with the change in the endogenous IAA level [58]. Cold stress leads to considerable changes in the levels of organic solutes and phytohormones in plant roots, which are crucial in cold stress responses. Such responses involve complex interactions between auxin signaling pathways and other regulatory mechanisms, which may cause different expressions of related genes, including *TrIAA27* in roots at specific times of interactions.

Drought stress upregulated the expression of the *IAA27* gene in alfalfa (*Medicago sativa*) [59]. Exogenous application of PoW (a harpin protein from *Ralstonia solanacearum*) alleviated drought effects in alfalfa; however, exogenous application of PoW coupled with drought stress did not change IAA27 expression significantly compared to the control, indicating negative control of PoW on *IAA27* expression in *M. sative* [59]. Zhang et al. (2020) [5] reported a higher relative expression of *IAA27* under drought stress, but this differential expression diminished when exogenous auxin and drought stress were applied together on white clover, indicating a repression effect of auxin on IAA27 under drought stress at least. In *Arabidopsis*, a component of RNA-directed DNA-methylation machinery *CLSY1* has been found mediating transcriptional repression of *AtIAA27*, thereby promoting lateral roots development in *Arabidopsis* under K deficient condition for sustaining growth in a low K stress condition [60], indicating DNA-methylation mediated regulation of IAA27 in the backup loop for post-translational protein degradation for maintaining the growth and development of plants in challenging environments. A mutation (C-T) in the coding region of apple *MdIAA27* improved the apple’s tolerance to phosphorus deficiency by growing longer and dense adventitious roots and absorbing higher phosphorus [54]. The MdIAA27 protein showed interaction with ARF8, ARF26, and ARF27, while a mutation in *MdIAA27* altered this interaction and promoted the release of ARFs, which upregulated *Small Auxin-Up RNA 76* (SAUR76) and *lateral organ boundaries domain 16 (MdLBD16)* [54], thereby enhancing root development in apple [61,62]. Priming of seeds with SA induced *IAA27* expression and improved seed germination, seedling vigor, and defense against *verticillium* wilt in Brinjal [63]. A recent study reported that the *Arabidopsis* gene *AtIAA27* interacted with tobacco mosaic virus (TMV) replicase, which resulted in disruption of nuclear localization of *AtIAA27* [64]; thus, biotic interactions may also disrupt *IAA27* functioning. In the current study, the longer and dense roots coupled with endogenous IAA interactions assisted the *TrIAA27* overexpressing plants to survive better in challenging environments.

Overexpression of *TrIAA27* in *Arabidopsis thaliana* resulted in increased growth in primary and secondary roots, above-ground biomass, plant height, and leaf chlorophyll contents, demonstrating that these traits are positively correlated with the higher level of TrIAA27 in *Arabidopsis*. The observations are in line with that which states that mutation in a single gene of the *Aux/IAA* family can affect growth and development in plants [22,23,24,25]. These observations are also partially in line with orthologs of *TrIAA27* in other species; for instance, overexpression of blueberry *IAA27* in *A. thaliana* resulted in more shoot branches [53], while silencing of *Sl-IAA27* in tomatoes reduced the chlorophyll contents in tomato leaves [49]. The overexpression of apple *MdIAA27* resulted in deeper apple roots with a higher number of adventitious roots under phosphorus-deficient conditions. *Ilex verticillata* genotypes with long stems and few branches had a higher expression of IAA27 and endogenous auxin content compared to a genotype with a short stem and more branches [65]. Rice cultivar (Jinhui 809) with a large panicle and higher seed setting percentage highly expressed OsIAA27 protein in flag leaves during the grain filling stages, thereby influencing photosynthesis transportation [66]. Mutation in soybean protein GmIAA27 resulted in the inhibition of apical dominance and promotion of lateral branch development [67]. Contrary to the current study observation, overexpression of *Sl-IAA27* reduced root length and lateral root formation in tomatoes [49]. Overexpression of blueberry *VcIAA27* in *A. thaliana* caused downward-curled leaf growth, a shorter sterile silique, and shorter plant stature [53].

Plants with height and biomass in wheat [68] and broader leaves in rice [69] performed better under drought stress compared to smaller and narrow-leaved plants. Exogenous application of IAA enhances alkaline stress tolerance in rice plants by regulating root development, reactive oxygen species scavenging, and gene expression related to IAA biosynthesis, transport, and catabolism [69]. *IAA27* has been reported to show pleiotropic effects in plants, and overexpression of *TrIAA27* in *A. thaliana* enhanced root development, above-ground biomass, and chlorophyll content in leaves. Generally, plants adjust their root architecture and chlorophyll contents in response to variable environmental cues, and changes in the root system, chlorophyll content, biomass, plant height, or leaf size may contribute to plant performance under stress [68,69,70]. Plants with efficient root systems can access deeper soil moisture, and higher chlorophyll levels ensure sustained photosynthesis, critical for plant growth and survival under drought stress [71]. *TrIAA27* is highly upregulated in response to the exogenous application of drought, salt, and ABA in the roots and leaves of white clover. Overexpression of *TrIAA27* had positive effects on the root system and chlorophyll content. The overexpression lines performed better under drought and salt stress conditions. All these observations are well in agreement with responses observed in other Aux/IAA family genes in different plant species. For instance, overexpression of rice *OsIAA18* improves root architecture and enhances salt and drought tolerance by upregulating ABA biosynthesis and signaling pathways [46]. *OsIAA6* of rice is involved in drought tolerance and tiller outgrowth, enhancing root development and maintaining higher chlorophyll content under drought stress conditions [72]. Overexpression of orchardgrass *DgIAA21* reduces drought tolerance by affecting root architecture negatively, decreasing total chlorophyll content, and increasing stress markers like MDA and electrolyte leakage [73]. Overexpression of rice *OsIAA20* enhances drought and salt tolerance by regulating ABA signaling and reducing stomatal closure, which in turn helps maintain higher proline and chlorophyll levels while reducing oxidative stress markers [47]. Conclusively, a better root system and higher chlorophyll content significantly improve drought and salt stress resistance in plants by enhancing water uptake, maintaining photosynthesis, and reducing oxidative damage. These adaptations are facilitated by various physiological and biochemical mechanisms, including improved antioxidant defense and enhanced water use efficiency.

The most obvious impact of environmental stresses (drought or salt) on plants is the inhibition of cellular or organ differentiation, resulting in plant growth, developmental restriction, and even plant death in severe cases. The damage caused to plants at the time of injury included lipid peroxidation, ion toxicity, and other secondary reactions induced by osmotic or oxidative stress and ion toxicity [55,74]. In the current study, the white clover gene *TrIAA27* showed significantly higher expression in response to drought, salt, NO, Ca^2+^, H_2_O_2_, and Spm in white clover (Figure 2). Overexpression of *TrIAA27* in *A. thaliana* improved drought and salt tolerance manifested by relatively less growth inhibition and physiological damage compared to the wild-type under stress conditions. Thereby, the overexpression lines had higher fresh and dry weights, survival rates, relative water content, photosynthesis performance indices on an absorption basis, Fm/Fv ratios, chlorophyll content, CAT (Ug^−1^), and POD (Ug^−1^) and lower wilting rates, electrolyte leakage, MDA content, and H_2_O_2_ content compared to the wild-type under drought or salt stress conditions (Figure 5, Figure 6 and Figure 7). These phenotypic and physiological indicators have widely been used to assess drought and salt stress resistance in different plant species, including *Arabidopsis* [55,75].

In addition to growth and development, Aux/IAA genes play crucial roles in regulating plant stress responses [76,77].

The positive regulation of drought and salt stress in *A. thaliana* by TrIAA27 is essentially in agreement with the responses of other Aux/IAA genes in different species [5,78,79]. Additionally, overexpression of rice *OsIAA18* [80] and grape *VvIAA18* in tobacco plants improved their salt tolerance [81]. Overexpression of *OsIAA6/OsIAA20* enhanced drought and salt stress in *O. sativa* [47,72]. In *A. thaliana*, mutations in *IAA19* genes result in reduced stress tolerance [76], while *AtIAA5*, *AtIAA6*, and *AtIAA19* were involved in drought stress responses and stomatal regulation [76]. *AUX/IAA* family members show varying numbers and functions in plant species; despite auxin, they can also interact with a variety of hormone-signaling molecules, including ethylene, brassinolide steroids, jasmonic acid, and salicylic acid [25,59,64,82], indicating complexity in dissecting their functions.

Decreased relative water content, higher electrical conductivity, and malondialdehyde content are the comprehensive reflections of plant response to salt or drought stress [55,74]. Under normal conditions, the production and elimination of free radicals and oxygen activities are in a dynamic balance. However, environmental stress can disrupt the dynamic balance and alter plants’ physiological and biochemical indexes [55,74]. In the current study, under drought and salt stress, *Arabidopsis* plants overexpressing *TrIAA27* had higher relative water content, CAT, and POD and lower electrical conductivity, MDA, and H_2_O_2_ content than the wild-type. Therefore, it is speculated that the *TrIAA27* gene is involved in scavenging reactive oxygen species (ROS), mitigating oxidative damage, and thereby alleviating drought and salt effects by producing dense and lengthy roots, probably absorbing more nutrients and water, improving photosynthesis ability, and adjusting endogenous hormones levels under challenging/stressed environments, which is in line with previous studies of *IAA27* in some plant species [54,60,63]. The current study only preliminarily confirmed the *Tr-IAA27* gene’s function and other family members’ functions, and the regulatory network of auxin-associated responses needs further in-depth mechanism-based investigations. The present study will serve as a foundation for future mechanism-based studies involving the Aux/IAA family and plant responses under salt or drought conditions. Overall, this study revealed that *Tr-IAA27* positively regulates plant growth and development and mitigates the effects of drought and salt stresses; therefore, it is a potential candidate to improve the biomass, drought, and salt tolerance of white clover.

## 4. Materials and Methods

### 4.1. Plant Materials and Growth Conditions

Uniform-sized healthy white clover seeds (cv. Landino) were surface sterilized with 0.5% (*w*/*v*) NaClO, sown on moisturized quartz sand, and placed under the light incubator for seven days for germination. Seedlings were then transferred to Hoagland’s nutrient solution for culturing for two weeks. Hoagland’s nutrient solution was replaced by every two days. The growth conditions were 12 h photoperiod, day/night temperatures of 23/19 °C, 75% relative humidity, and 250 μmol m^−2^⋅s^−1^ photosynthetic photon flux density. After 22 days, white clover plants were subjected to exogenous abiotic, signaling molecules, and phytohormone treatments.

*A. thaliana* ecotype Col-0 was used for the genetic transformation. Seeds of wild-type Col-0 and overexpression lines were surface-sterilized with alcohol (75%, *w*/*v*) and NaClO (0.1%, *w*/*v*) and then sown on Petri dishes containing ½ MS media supplemented with sucrose (3%) and agar (0.7%). Petri dishes were placed at 4 °C in the dark for 2–3 days for vernalization, then transferred to the growth chamber in the set condition of 21 °C with 65% relative humidity, 16 h photoperiod, and 150 μmol m^−2^⋅s^−1^ photoactive radiation. After 5 days, uniform-sized seedlings were transferred to plates containing ½ MS medium, sucrose (3%), and agar (0.7%) supplemented with or without mannitol (100 and 200 mML^−1^) and NaCl (100 and 150 mML^−1^). Plates were arranged at 90 degrees in the growth chamber to observe differences in roots and shoot growth under optimal, drought, or salt stress conditions between overexpression and wild-type *A. thaliana* plants. Additionally, 10-day-old well-grown, uniform-sized wild and transgenic *A. thaliana* plants were transplanted into plastic pots containing nutrient soil. Pots were placed in the growth chamber to raise the plants for an additional three weeks and then subjected to different treatments at about 25 d of transplanting. For different kinds of physiological and genetic analysis, the samples were collected from white clover and *A. thaliana* plants at specific growth stages (details below), immersed immediately in liquid nitrogen, and stored at −80 °C for further use.

### 4.2. Exogenous Treatment of Heavy Metal, Abiotic Stress, Signalling Molecules, and Phytohormones to White Clover and Sample Collection

For heavy metal, salt, and drought stress treatments on white clover, seedlings were immersed in the solution containing CdSO_4_ (600 μM), NaCl (200 mM), and 15% PEG6000 *w*/*v*, respectively. For high- and low-temperature stress treatments, seedlings were transferred into growth chambers adjusted to 35 °C and 4 °C, respectively. For signaling molecules and phytohormones, the plantlets were dipped into solutions containing NO (SNP 25 μM), H_2_O_2_ (10 mM), Spm (20 mM), Ca^2+^ (CaCl_2_ 5 mM), ABA (100 μM), and IAA (1 μM), respectively. The roots and leaves of 15 plants with three biological replicates were collected after 0, 1.5, 3, 6, 12, and 24 h of treatments. Each plant’s second mature leaf and ¾ roots were collected and immediately frozen in liquid nitrogen. Finally, samples were stored at −80 °C for subsequent RNA extraction and further use.

### 4.3. Drought and Salt Stress Treatments in Arabidopsis

The drought and salt stress treatments were according to Jia et al. (2021) [55]. Briefly, homozygous T3 overexpression lines and wild-type *A. thaliana* plants were subjected to drought and salt stress treatments. Five days old plants of overexpression and wild-type lines were transferred onto ½ MS medium (15 plants of each genotype with three technical repeats) supplemented with 100 mML^−1^ or 200 mML^−1^ mannitol for drought stress and 100 mML^−1^ and 150 mML^−1^ of sodium chloride for salt stress treatments. The phenotypes of plants were observed daily for up to 26 days. Photographs of the plants were taken using a digital camera. The yellowing rate (senescence) was calculated by counting the respective plant at the end of the experiment from each treatment. One-month-old plants of overexpression and wild-type *A. thaliana* lines were also subjected to drought and salt stress treatment in nutrient soil (peat moss, vermiculite, perlite, 3:1:1) under controlled growth chamber conditions. Drought stress treatment was induced by withholding the irrigation for 22 days. Salt treatment was applied by irrigation with 100 mML^−1^, 200 mML^−1^, and 300 mML^−1^ of NaCl every two times after every two consecutive days. Photographs were captured when the phenotypic differences were obvious, and the survival rates and fresh and dry weights were calculated at the end of the experiment.

### 4.4. Isolation of the IAA27 Gene from White Clover

The full-length coding sequence of *TrIAA27* was isolated from the cDNA of white clover by SMARTer ^®^ Race 5′ and 3′ kit (Clonetech Laboratories, Mountain View, CA, USA). The conserved sequence regions of *IAA27* genes of *Medicago truncatula*, *Cicer arietinum*, and *Glycine max* were identified by the BLAST function of NCBI (http://www.ncbi.nlm.nih.gov/ (accessed on 12 March 2021)). Premier primer 5 soft was used to design primer pairs (TrIAA27-F and TrIAA27-R, Appendix A). The first fragment of the *TrIAA27* gene was obtained by RACE PCR. The reaction system for PCR consisted of 1 μL cDNA, 1 μL each primer, 25 μL TaKaRa PrimeSTAR^®^ Max DNA Polymerase, and 22 μL dd H_2_O. The specific reaction conditions were denaturation at 94 °C for 5 min, followed by 35 cycles of denaturation at 98 °C for 10 s, annealing at 58 °C for 5 s, extension at 72 °C for 5 s, and then final extension at 72 °C for 10 min. The amplified fragment was cloned into pMD™ 19-T vector for sequencing by following the cloning kit protocol. After confirming the accuracy of the cloned fragment by NCBI-blast function, two gene-specific RACE primers (TrIAA27-RACE3′ and TrIAA27-RACE5, Appendix A) were designed for amplifying both ends of the *TrIAA27* gene. The reaction system and PCR conditions were adjusted according to the instructions of SMARTer^®^ RACE 5 and 3′Kit’s manual.

### 4.5. Bioinformatics Analysis

The characteristics of the protein of *TrIAA27*, including PI value, molecular weight, and net charge, were determined by ExPASy (http://web.expasy.org/protparam (accessed on 21 May 2021)). Clustalx and MEGA11 were used for multiple sequence alignment and phylogenetic tree construction, respectively. PSORT was used to predict subcellular localization of TrIAA27 protein.

### 4.6. Total RNA Isolation, Construction of cDNA, and qRT-PCR Analysis

Tissue samples of white clover and *A. thaliana* were grounded in liquid nitrogen. A total of 100 mg of tissue powder was used to extract the total RNA of each sample using HiPure Universal RNA Mini Kit (Magen, Beijing, China) following the manual’s instructions. After confirming the quantity and quality of total RNA using a NanDrop spectrophotometer and RNA integrity using 1.5 agarose gel, the complementary DNA (cDNA) was synthesized by reverse transcription of the first strand of RNA using MonScript™ RTIII All-in-One Mix with dsDNase (Monad Biotech Co., Ltd., Beijing, China) following kit protocols. Relative gene expression was determined using 2X M5 HiPer SYBR Premix EsTaq (with TliRNaseH) (Mei5 Biotechnology, Co., Ltd., Beijing, China) using CFX96 Real-Time PCR detection system (Bio-RAD, Hercules, CA, USA) according to manual’s instructions. The qRT-PCR primers are provided in Appendix A. The TrB-Actin gene was used as an internal control for white clover. The amplification conditions were as follows: initial denaturation at 95 °C for 5 min, then 40 cycles at 95 °C for 10 s and 57 °C for 30 s. The relative difference in the expression of the genes was calculated with the 2^−ΔΔCT^ method [83].

### 4.7. Plasmid Construction and Genetic Transformation

pSUPER1300-GFP and pCAMBIA1301 vectors were used for subcellular localization and constructing overexpression *A. thaliana* lines, respectively. The complete coding region (without stop codon) of *TrIAA27* was amplified from previously constructed pMD19 simple vector by PCR reaction using PrimeSTAR^®^ Max DNA Polymerase (Takara-Bio, Kusatsu, Japan) and inserted into *HandIII* and *EcoRI* site of pCAMBIA1301 (containing CaMV 35S promoter). *XbaI* and *KpnI* sites of the pSuper1300 vector were used to generate *TrOXI1*::GFP in-frame fusion protein using the Easy Geno Assembly Cloning kit (TIANGEN BIOTECH (BEIJING) Co., Ltd, Beijing, China). The primers for amplifying coding region and flanking vector sequences for both constructs are enlisted in Appendix A. The constructed pCAMBIA1301 vector (CaMV 35S::TrIAA27) was then mobilized into *A. thaliana* (Col-0) by *Agrobacterium*-mediated transformation through the floral dip method followed by Clough and Bent. (1998) [84]. The seeds of transformed plants were harvested, surface-sterilized, and sown on selective media of ½ MS media with hygromycin-50. The surviving plants were transferred into the soil at the four-leaf stage. The *TrIAA27* transgenic lines were confirmed by PCR. Six independent transgenic overexpression lines were developed. The expression level of *TrIAA27* in transgenic lines was assessed by qRT-PCR. The overexpression lines OE3 and OE5 were selected for further experiments based on their higher relative expression levels.

### 4.8. Subcellular Localization

The fusion constructs *TrIAA27*::GFP and control GFP vectors were inserted into the *Agrobacterium* strain EHA105 competent cells using a standard protocol. Briefly, the transformed *Agrobacterium* culture was shaken for 2–3 h in YEB (Yeast Extract Beef) solution without an antibiotic at 28 °C, then centrifuged at 4000 rpm for one min (minute) and plated 100 µL on YEB medium containing rifamycin-20 and kanamycin-50, and incubated at 28 °C for 72 h, followed by positive colonies selection using colony PCR. Cell culture with positive colonies was shaken at 28 °C until the OD-600 value reached 0.8, then the 30 mL cell culture was centrifuged at 4000 rpm for 10 min at 4 °C, dissolved in autoclaved water containing 5% sucrose and 0.02% silwet, and, finally, introduced into tobacco leaves. The subcellular location of the *TrIAA27* protein was determined by transiently expressing the TrIAA27::GFP fusion protein in the leaves of *Nicotiana benthamiana*. Tobacco plants were placed in a growth chamber in the dark, set at a day/night temperature of 23/19 °C, with 75% relative humidity for 24 h, followed by observing the GFP localization using a fluorescent microscope (Olympus, Tokyo, Japan).

### 4.9. Determination of Relative Water Contents (RWC)

RWC was measured as previously described by Barrs and Weatherley [85]. A total of 0.3 g of leaf blade samples was collected between 9:00–10:00 a.m., wrapped well in ordinary absorbent paper, and placed into centrifuge tubes of 50 mL, followed by filling the tubes with water, covering with a lid, and placing in a protected place for 24 h. When the leaves had absorbed water at full saturation level, leaf samples were removed, wiped off their surface, weighed the saturated fresh weight, and placed the samples in a blast oven set at 105 °C temperature for 45 min. Samples were dried at 75 °C to reach a constant weight and finally measured their dry weights. Three biological repeats were used for all treatments. RWC of the leaves was calculated by employing the following formula:RWC % = [(FW − DW)/(TW − DW)] × 100 

FW, TW, and DW represent fresh weight, saturated fresh weight after drenching the leaves in water for 12 h, and dry weight, respectively.

### 4.10. Determination of Relative Electrical Conductivity (EL)

Relative electrical conductivity was determined as previously described by Blum and Ebercon [86]. Briefly, 0.1 g of leaf blade samples of the different materials in the study was placed into test tubes filled with 15 mL of deionized water. The initial conductivity (C_initial_) was determined by the conductivity meter Model 32 (YSI, Yellow Springs, OH, USA) after 24 h, then we boiled the samples at 100 centigrade for 15 min, followed by cooling down at room temperature and measuring their final conductivity (C_max_). Three biological repeats were used for all treatments. The following formula was used to calculate EL:Relative electrical conductivity EL = C_initial_/C_max_ × 100 
where C_initial_ is initial conductivity, and C_max_ is final conductivity.

### 4.11. Determination of Chlorophyll Fluorescence Parameters

For determining chlorophyll fluorescence parameters, we placed the soil-grown WT and transgenic lines in the dark for 30 min and measured the maximum quantum yield of PSII photochemistry (Fv/Fm) and PI_ABS_ with a pulse-amplitude modulation portable chlorophyll fluorometer (PAM-2500, Walz, Effeltrich, Germany), using 15 replicates for each treatment.

### 4.12. Determination of Malondialdehyde (MDA) Content

MDA contents were determined according to Heath and Packer’s (1968) [87]. Briefly, 0.1 g leaf tissues was immediately frozen in liquid nitrogen, followed by thoroughly being grounded in ice within 2 mL of 50 mM pre-cooled phosphate-buffered saline (PBS) solution (pH 7.8) and centrifuged at 12,000× *g* for up to 30 min at 4 °C. The supernatant was taken and used to extract and determine MDA content by adding 1 mL of reaction solution comprising trichloroacetic acid (20% *w*/*v*) and thiobarbituric acid (0.5% *w*/*v*) of a 0.5 mL solution of crude enzyme. Then, samples were heated at 95 degrees Celsius for up to 30 min in a water bath, followed by quickly cooling to room temperature in an ice bath, with continuous gentle shaking to avoid bubble formations, and then centrifuged at 10,000× *g* for 10 min, removing the air bubbles, if any, with the pipette. Finally, we took out the supernatant and measured absorbance values at 532 nm and 600 nm wavelengths. Three biological repeats were used for all treatments. To calculate MDA contents values, we subtracted the absorbance value of 600 nm wavelength from the absorbance value obtained at 532 nm wavelength by using an extinction coefficient of 155 mM^−1^ em^−1^ [87].
MDA (nmole g^−1^ DW) = (A532 − A600) × V × 1000/155 × W 
where A532 is the absorbance at 532 nm, A600 is the absorbance at 600 nm, V is the extraction volume, and W is the dry weight of the leaf.

### 4.13. Measuring H_2_O_2_, CAT, and POD

The concentration of H_2_O_2_ was determined with the updated (NH4)_2_Fe(SO_4_)/Xylenol orange procedure [88], and the catalase (CAT) activity was determined by calculating the concentration level reduction of H_2_O_2_ [75]. The peroxide activity (POD) was determined with the substrate 4-methyl catechol to test the POD behavior [89].

### 4.14. Statistical Analysis

IBM SPSS Statistics 21 software was used for statistical analyses, and GraphPad Prism 8.3.0 was used to draw graphs.

## 5. Conclusions

In the current study, we systematically explored the function of the Aux/IAA family member *Tr-IAA27* of white clover in response to drought and salt stress. The study results indicated that *Tr-IAA27* was strongly induced in response to exogenously applied drought, salt, IAA, and ABA in leaves of white clover. In addition, overexpression of *Tr-IAA27* in *A. thaliana* increased plant size and chlorophyll content and improved the drought and salt tolerance in transgenic *Arabidopsis* plants by promoting photosynthesis rate, ROS scavenging ability, antioxidant defense, plant survival, and reducing membrane damage, lipids peroxidation, relative water loss relative to wild-type control. Overall, the results of this study confirmed that *Tr-IAA27* can improve the drought and salt tolerance of plants and is more strongly expressed in leaves than roots of white clover in response to drought, salts, heavy metals, IAA, and ABA. Overexpression of Tr-IAA27 in white clover can improve biomass accumulation and the drought and salt tolerance of white clover; thus, it could be considered as a potential target gene for further use, including the breeding of white clover for higher biomass with improved root architecture and tolerance to abiotic stress.

## Figures and Tables

**Figure 1 plants-13-02684-f001:**
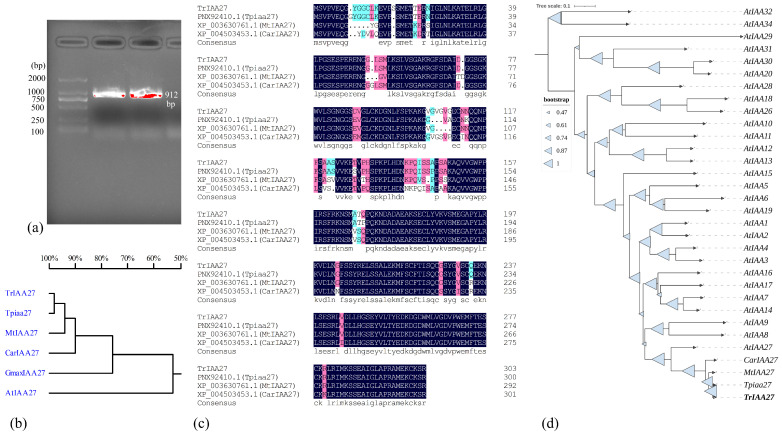
The gene sequence characteristics and phylogenetic analysis of the *Tr-IAA27* gene of white clover. (**a**) Amplified band of *Tr-IAA27* from white clover cDNA; (**b**) homology of nucleotides bases of *Tr-IAA27* with closely related species; (**c**) comparison of amino acid sequences of *Tr-IAA27* with other related *IAA27* proteins, highly homolog amino acid residues are shaded; (**d**) phylogenetic relationship of *Tr-IAA27* proteins with same protein from closely related species and *Arabidopsis thaliana*; triangles in branches represent bootstrap values, and the phylogenetic tree analysis was based on minimum evolution using Mega (version 11).

**Figure 2 plants-13-02684-f002:**
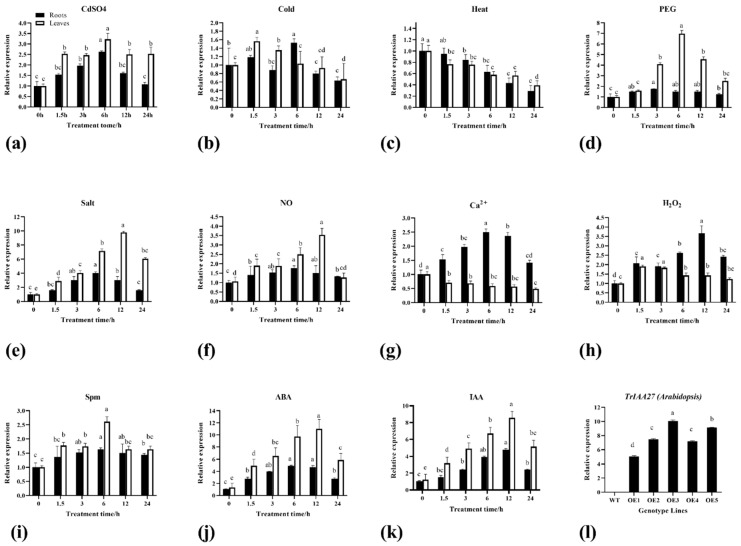
Relative expression of *Tr-IAA27* in white clover in response to different external stimuli. (**a**–**k**) Show relative expression of *TrIAA27* gene in response to treatments of heavy metal (600 mM CdSO_4_), cold (4 °C), heat (35 °C), drought (15% PEG6000 *w*/*v*), salt (200 mM NaCl), NO (25 μM), Ca^2+^ (5 mM), H_2_O_2_ (10 mM), Spm (20 mM), ABA (100 μM), and IAA (1 mM), respectively. (**l**) Expression of *TrIAA27* in wild-type *Arabidopsis* and 5 overexpression lines (OE1-5) of *Arabidopsis*. Different alphabet letters over bars show statistically significant differences (ANOVA, *p* < 0.05), and the error bar shows standard error (SE).

**Figure 3 plants-13-02684-f003:**
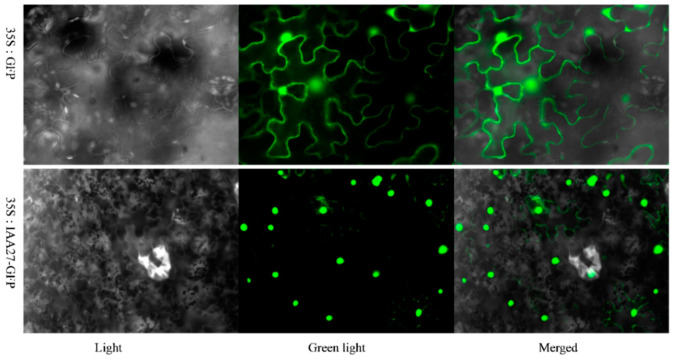
Subcellular localization of *TrIAA27* in tobacco leaves. The TrIAA27::GFP vector and an empty vector were transferred into tobacco leaves for transient expression and observed under the fluorescent microscope. The green color shows signals of the *TrIAA27* protein.

**Figure 4 plants-13-02684-f004:**
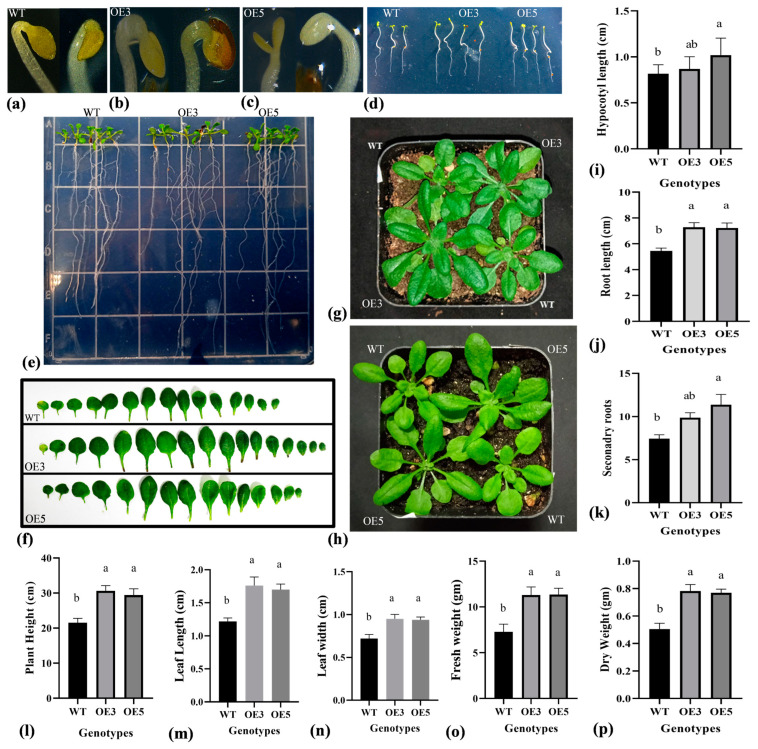
Phenotypic comparison of growth performance indicators between wild-type (WT) and *TrIAA27* overexpression *Arabidopsis* lines (OE3 and OE5). (**a**–**c**) Display hook angles of WT, OE3, and OE5, respectively. (**d**,**i**) Show a comparison of hypocotyl length between WT, OE3, and OE5. (**e**,**j**,**k**) Roots growth comparison of WT, OE3, and OE5 grown on ½ MS medium supplemented with 3% sucrose and 0.7% agar. (**f**,**m**,**n**) Leaf length and width differences among WT, OE3, and OE5. (**g**,**h**,**i**,**l**,**o**,**p**) Growth performance, hypocotyl length, plant height, fresh weight, and dry weight of WT, OE3, and OE5, respectively. Different alphabet letters over bars show statistically significant differences (ANOVA, *p* < 0.05) among genotypes, and the error bar shows standard error (SE). WT, OE3, and OE5 represent wild types, overexpression lines 3 and 5, respectively.

**Figure 5 plants-13-02684-f005:**
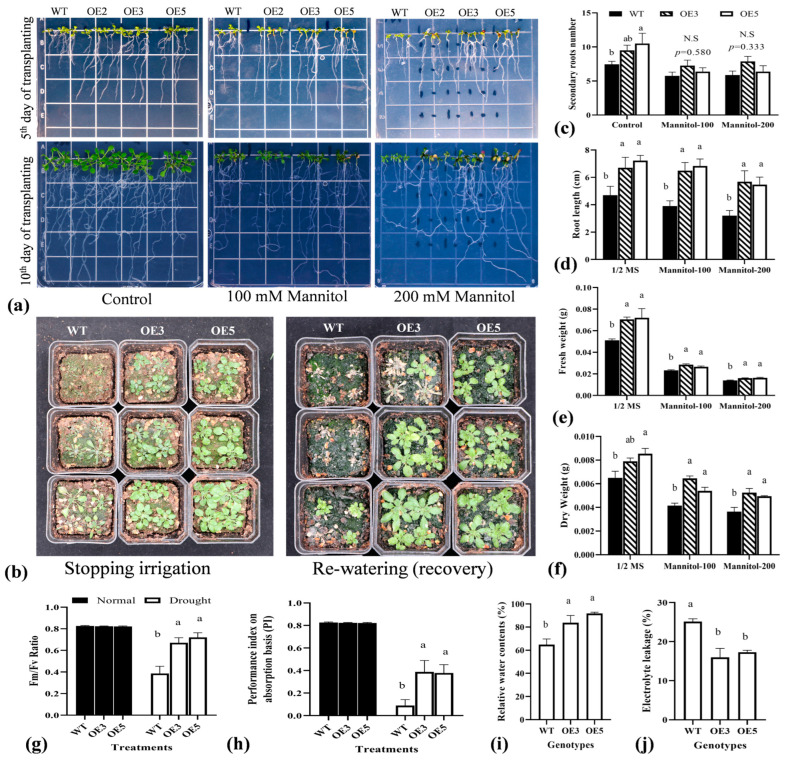
Growth performance of wild-type (WT) and *TrIAA27* overexpression lines (OE3 and OE5) under drought stress conditions. (**a**) Phenotypical comparison of WT, OE3, and OE5 grown on ½ MS media containing 3% sucrose and 0.7% agar supplemented with 0 mM mannitol, 100 mM mannitol, or 200 mM mannitol on 5th and 10th d. (**b**) Phenotypes of WT, OE3, and OE5 grown in soil after irrigation withholding at 15 d. (**c**–**j**) Display statistical comparisons of secondary roots no., roots length, fresh weights, dry weights, photochemical efficiency (Fm/Fv ratio), performance index on an absorption basis (PI), relative water content %, and electrolyte leakage %, respectively. Different alphabet letters over bars show statistically significant differences (ANOVA, *p* < 0.05) among genotypes; N.S represents statistically non-significant differences. The error bar shows standard error (SE). WT, OE3, and OE5 represent wild types, overexpression line 3 and overexpression line 5, respectively.

**Figure 6 plants-13-02684-f006:**
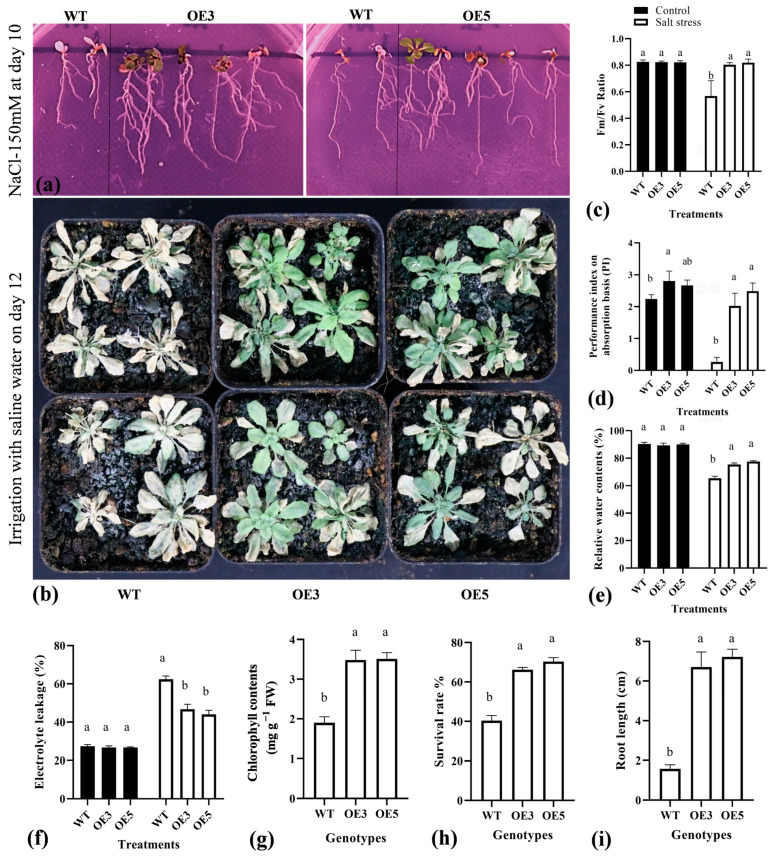
Growth performance comparison of wild-type (WT) and *TrIAA27* overexpression (OE) *Arabidopsis* lines under salt stress conditions. (**a**) Performance of WT, OE3, and OE5 on ½ MS media supplemented with 150 mM NaCl at day 10. (**b**) Performance of WT, OE3, and OE5 in soil irrigated with saline water. (**c**–**i**) Statistical comparisons of photochemical efficiency Fm/Fv, performance index on an absorption basis, relative water content %, electrolyte leakage %, chlorophyll contents, survival rates, and root length, respectively. Different alphabet letters over bars show statistically significant differences (ANOVA, *p* < 0.05) among genotypes, and the error bar shows standard error (SE). WT, OE3, and OE5 represent wild types, overexpression line 3, and overexpression line 5, respectively.

**Figure 7 plants-13-02684-f007:**
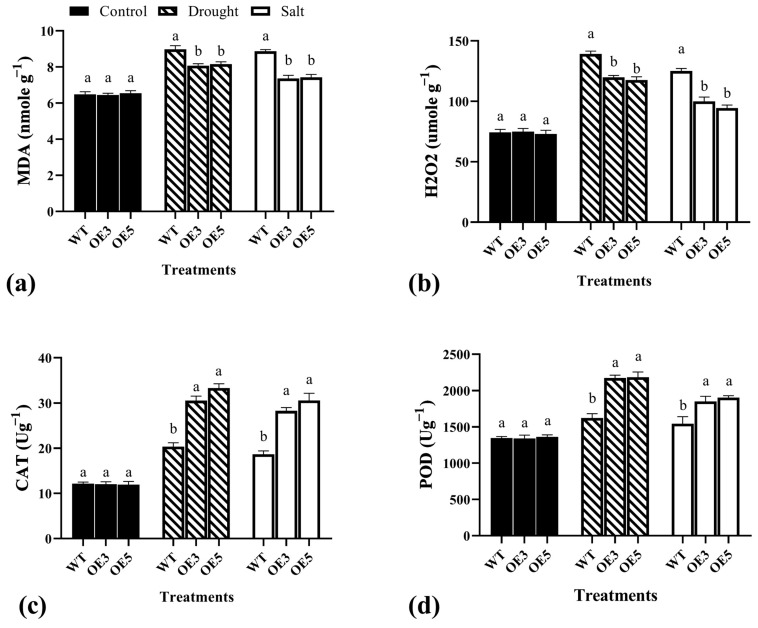
The comparative analysis of endogenous oxidants and antioxidant levels in the wild-type and overexpression *TrIAA27 Arabidopsis* under drought and salt stress conditions. (**a**) Relative malondialdehyde (MDA) contents, (**b**) relative hydrogen peroxide (H_2_O_2_) concentration, (**c**) endogenous catalase (CAT), and (**d**) the peroxidase (POD) concentration in the leaves of wild-type and *TrIAA27*-overexpressing *Arabidopsis*. Different alphabet letters over bars show statistically significant differences (ANOVA, *p* < 0.05) among genotypes, and the error bar shows standard error (SE). WT, OE3, and OE5 represent wild types, overexpression line 3, and overexpression line 5, respectively.

## Data Availability

All data supporting the conclusions of this study have been included in the original manuscript or Appendix A and can further be requested from the corresponding author.

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
