# Peer review of "Overexpression of Auxin/Indole-3-Acetic Acid Gene TrIAA27 Enhances Biomass, Drought, and Salt Tolerance in Arabidopsis thaliana"

_plants, 2024, doi:10.3390/plants13192684_

Round 1
Reviewer 1 Report
Comments and Suggestions for Authors
Please find my comments attached.

Moderate editing of the English language is required.
Author Response
Thank you very much for taking the time to review this manuscript. Please find the detailed responses in the attachment. The corresponding revisions/corrections are highlighted red in re-submitted files.
Best Regards,
Muhammad Zafar Iqbal

Reviewer 2 Report
Comments and Suggestions for Authors
My comments and suggestions are attached.

Minor editing is required.
Author Response

(The authors gave the same response as above.)

Reviewer 3 Report
Comments and Suggestions for Authors
The manuscript by Iqbal et al. describes the effects of various abiotic stresses on the expression of the auxin response gene TrIAA27. Furthermore, cloning and over expressing this gene in Arabidopsis resulted in various increased parameters including reduced wilting rates, lowered reactive oxygen species (although only one was measured), and electrolyte leakage, and in increased biomass, chlorophyll content and others. I have very little issues with the general concept of the manuscript and most of its organization. However the manuscript requires extensive language editing. Often, articles and helping verbs are missing, and sentences are also often too long, so that it is difficult to follow. While this can easily be fixed, other issues may need some further explanation. First, why was 1mM of auxin used? This is a very high concentration that is probably close to being a herbicide. Usually, low micro molar concentrations are used for experiments. This needs an explanation and a justification. Also, why was Calcium used? It is usually regulated by ion channel activities, but not so much through exogenous application. I appears as if these treatments were applied by dipping the plants in a solution of these compounds. Why was this method used? And how long was the dipping? Section 1.8 to 1.11 all followed an Iqbal paper. This can either be combined, or at least a brief description should be provide, which I would prefer since it gives the reader the relevant information right away. Under 2.2 (results) there are several point that need revision. In the first paragraph there it says "highest under drought" and then it say right after that "significantly higher". If it is highest, then the next one cannot be higher. The sentence needs rewording. At the end of the same paragraph several fold increases are listed without giving the reader the time points, which should be added. In the next paragraph (still 2.2) it says " under all observed time spans". However, only time points are listed. This needs to be changed.
A more general question is: does Arabidopsis have a homolog to TrIAA27? This is relevant in evaluating the results presented here. If so, is a mutant available for that gene? If so, It would have been better to use that so that there is no background interference.
In the first paragraph of 2.4, it says "under 100mM and 200mM stressed conditions". This should be reworded for clarification.
The last sentence of the conclusion ("it is a vital genetic resource") is not clear to me as to why it is vital. This could only be stated if a knock-out would proof this.
In the figures, the time point when measurements were done should be added. Also, if only one time point was selected, what is the justification?
Figure 5 A has no ANOVA.
Comments on the Quality of English LanguageAlready listed above. I recommend to revise the whole manuscript with regard to grammar in particular by having a native speaker involved.
Author Response

(The authors gave the same response as above.)
